# Trees and Crops Arrangement in the Agroforestry System Based on Slope Units to Control Landslide Reactivation on Volcanic Foot Slopes in Java, Indonesia

**Rina Purwaningsih** [1] , **Junun Sartohadi** [2,*] **and Muhammad Anggri Setiawan** [3]

1   Geo-Information for Spatial Planning and Disaster Risk Management, Post Graduate School, Universitas Gadjah Mada, Yogyakarta 55281, Indonesia; rinapurwaningsih@mail.ugm.ac.id
2   Department of Soil Science, Faculty of Agriculture, Universitas Gadjah Mada, Yogyakarta 55281, Indonesia
3   Department of Environmental Geography, Faculty of Geography, Universitas Gadjah Mada, Yogyakarta 55281, Indonesia; anggri@ugm.ac.id
*   Correspondence: junun@ugm.ac.id

**Abstract:** Agroforestry, as the dominant land use at the volcanic foot slope in Java Island, is prone to landslide due to a combination of rough relief and thick soil layer. However, evaluations of specific vegetation patterns against landslide reactivation due to soil erosion, which relays on the existing slope units and geomorphological processes, are still limited. The research data were collected through aerial photo interpretation by delineating morphological units of old landslides, slope units, and the existing land use. This was followed by field surveys for two consecutive purposes, i.e., (1) verification of aerial photo interpretation and (2) identification and intensity assessment of existing geomorphological processes. The data were tabulated according to slope units, as a basis for tree and crop arrangement in controlling erosion and landslide, by considering economic, social, and ecological functions. The agroforestry would control the landslides reactivation if the tree and crop arrangement was based on the morphological units formed by the previous landslide. The slope units are classified into residual zones at the highest elevations with flat slopes, erosion zones with the steepest slope, and sedimentation zones at the lowest elevations with gentle slopes. Trees and crops at those three units of the former landslide have different functions in controlling processes of rill erosion, gully erosion, and soil creep.

**Keywords:** agroforestry; crops; landslide; re-activation; slope; trees

## 1. Introduction

Landslides that occur at the volcanic foot slope and thick soil, especially in Java, leave a mark in the form of typical surface morphology. The landslides gravitationally redistribute soil mass and change the position of soil layers, which generally have different characteristics and have different potential uses [1–5]. The results of soil redistribution by landslides can be observed in three slope zones from the highest elevation to the lowest elevation, namely residual, erosion, and deposition [6–8].

Volcanic ash deposition, which results from eruption processes during the volcanic body formation period, forms a very thick soil at the volcanic foot slope zone. Very thick soils combined with the high elevation, and the steep slope angles make the volcanic foot slope prone to landslides. On the contrary, the volcanic foot slope has a great potential for agricultural sector development, especially in the former landslide area, where the soils have favorable characteristics to support plant growth [8]. One of the problems that might limit the agricultural sector development at the volcanic foot slope zone

is a limitation on irrigation water availability, thus rain-fed agriculture is widely applied in the area. Therefore, appropriate land management techniques should be applied to support the agricultural production in the landslide-prone area, such as volcanic foot slope [9]. The agroforestry system can be an option to optimize the agricultural sector development on dry land due to its function in controlling slope stability [10,11].

Agroforestry systems have become a common type of land use, as a compromise solution between natural forests that function ecologically and intensive use of land that functions economically. In Java, agroforestry is the most dominant land use (Figure 1), which has an economic and ecological function for soil and water conservation [12–14]. Thus, the traditional community needs agroforestry systems, such as the vegetative-based soil and water conservation techniques. However, during the agroforestry cultivation, the local community tends to do vegetation planting trial and error that is only based on economic value without considering the soil and water conservation [15–17]. Therefore, the ecological function of the agroforestry should be evaluated so that the effectiveness of soil erosion control and landslide reactivation control can be achieved [18–20].

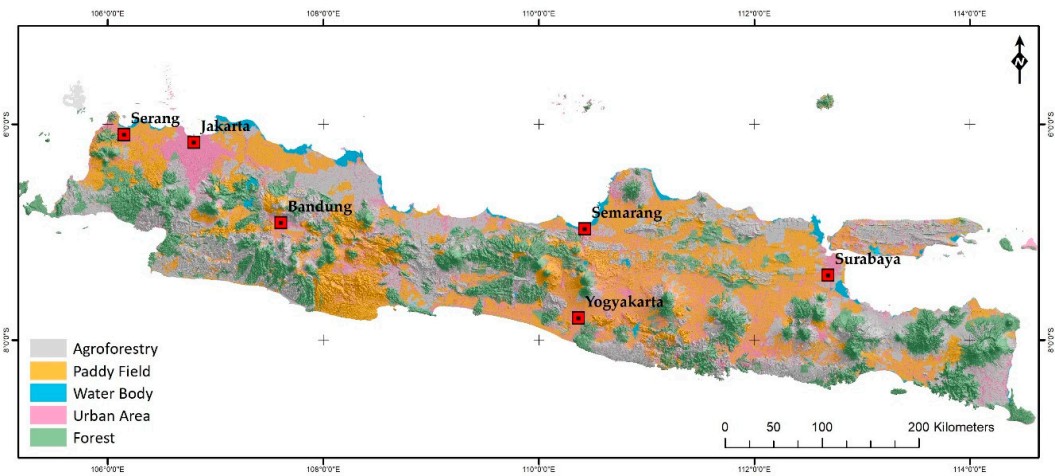

**Figure 1.** Rain-fed agriculture in the form of agroforestry and cassava field is extensive land use type in. Java Island. (Source: Reclassification of land cover information on RBI Map scale of 1:25,000 from BIG).

Landslide at very thick soil creates a unique land surface morphology which consists of three slope units, namely residual, erosional, and depositional slopes. The geomorphological processes at those sequential slope units are mainly soil erosion, that would initiate landslide re-activation. The soil erosion processes can be controlled by cultivating specific vegetation according to the slope units. In general, agroforestry systems consist of several types of trees and crops, yet the consideration of the vegetation arrangement based on the sequential slope units is still rarely discussed. Besides this, studies examining the vegetation arrangement by considering geomorphological processes type and intensity in landslide areas are still limited. Studies about the detailed arrangement of dryland plant species in the agroforestry system still have a limited purpose, to increase slope stability only, but do not pay attention to its economic and social functions [21–23]. Besides, plant layout arrangements that have also been widely discussed aim to protect the soil from nutrient loss due to erosion [24–26].

This study aimed to utilize slope zones of former landslides, which are determined by type and intensity distributions of geomorphological processes, as references for the site arrangement of plant species in an agroforestry system. The vegetation arrangement considers the types of local plants around the study site so that they can be ascertained to be ecologically suitable and can be accepted by the community.

The study was conducted on an erosion valley side undergoing a deepening process at a former landslide area of volcanic foot slopes. The deepening process of erosion valley occurred due to material removal of very thick soils, followed by geomorphological processes in the volcanic foot slope. The very

thick soils originate from ash deposits of recent volcanoes that are overlaid and developed on the previous volcano's body [27], then form a prominent cone morphology (Figure 2) [28,29].

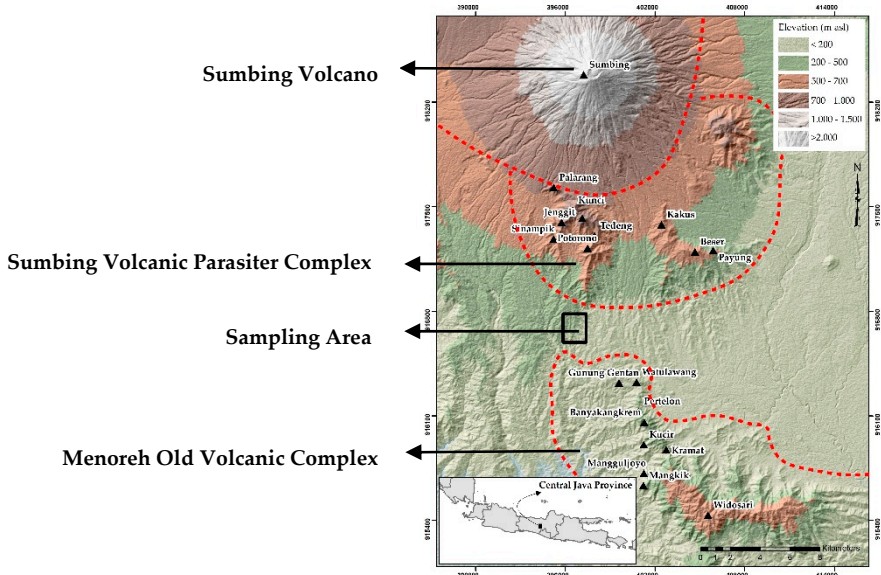

**Figure 2.** The research sampling location formerly experiencing landslide on the side of the erosion valley on the foot slope of Sumbing Volcano that is a transition area with the Menoreh Volcano complex.

## 2. Materials and Methods

The research sample site was located in Java Island, specifically on the foot slope of Sumbing Volcano, which is a transition zone to Mount Menoreh (Figure 2). In general, the research site is involved with Java physiology, that is, a volcanic island formed by tectonic plates' subduction. The Java Island is divided into three main physiographic units, namely the southern zone (raised block forming plateau morphology), the middle zone (volcanoes), and the northern zone (fold mountains) [30]. The southern zone is composed of old volcanic sediment material that has undergone various sinking and lifting processes interspersed with the deposition of marine facies rocks. While the middle zone is the current volcanic zone, mostly classified as active volcanoes by the Indonesian Center for Volcanology and Geological Disaster Mitigation (PVMG) [27]. On the other hand, the northern zone is a fold mountains zone with shallow marine facies rock material. The research sample site is located in the area between the southern and the middle zones of Java Island, classified as volcanic transitional zones where most of the large landslides occurred. In the transitional zone, the young volcanic materials overlay on the top of old weathered volcanic material, which generates areas with very thick soils.

Research data were collected in two ways, namely (1) visual and digital interpretation of aerial photographs and (2) field surveys. The visual interpretation was intended for slope unit delineations, geomorphological process identifications, and plant type identifications. Meanwhile, digital aerial photography interpretation was intended to measure the length and area dimension of geomorphological processes, plant height, plant canopy diameter, and plant density. Furthermore, the results of aerial photograph interpretation were verified in the field along with type and intensity assessment of geomorphological processes, types of plant, plant height, plant diameters, and plant density.

The aerial photograph is generated by the Unmanned Aerial Vehicle (UAV) technique, which resulted in 0.05 m of orthophoto resolution, 0.20 m of Digital Surface Model (DSM) resolution, and 0.564 m of Digital Terrain Model (DTM) resolution. The photos were taken during a dry session in August 2019. The orthophoto is printed out in scale of 1:2000 as field tool to identify the geomorphological processes and plant types.

The geomorphological processes' identification from aerial photographs interpretation was done in five stages, including the identification of gully erosion features that are easily recognizable due to the large dimension size, the identification of potential gully erosion used to determine rill erosion or gully erosion branches, and the identification of a landslide reactivation feature. Furthermore, the features found from the interpretation of aerial photographs of the cassava field were used to track the presence of sheet erosion, splash erosion, soil cracks, material displacement, and soil creep through field surveys. Meanwhile, the geomorphological process identification in the agroforestry system was directly identified by field surveys since they were not visible from aerial photographs.

Plant type identification of trees as perennial plant and crops as a seasonal plant included two stages from the aerial photograph interpretation and one stage from field surveys. The three stages included: (1) groups of trees that have prominent features such as height, canopy diameter, and density, (2) groups of crops in the cassava field, which are possibly located between the trees and (3) groups of intercropped plants between the trees, which can only be identified directly in the field.

All plant types in the sample site were inventoried, grouped, and arranged in each morphological unit based on considerations of plant morphology, ecological functions, and economic functions. A list of citations obtained through a literature review about the type of plant and its role in controlling geomorphological processes is presented in the discussion section.

In this research, local slope arrangement was applied as a basis for plant management in the agroforestry systems. Plants' arrangement is based on the global theory in geomorphology that every feature of earth surface is unique and appears in a sequential morphology, with materials, and processes of modifying morphology through time [31–33]. The application of that global theory in the local scale of land parcels is the slope arrangement covering the local highest to the lowest elevations. The effectiveness of the proposed plant re-arrangement might suppress the intensity of soil erosion processes that activate the landslide. The assessment of the effectiveness of the proposed vegetation arrangement would be based on the soil erosion characteristics that occur in every slope unit.

## 3. Results

### 3.1. Zones of the Landslide Slope Units

The results of aerial photo interpretation are sequential slope units including residual, erosional, and depositional zones, where different types and distributions of geomorphological processes are developed (Figure 3a). In the cassava field, the geomorphological processes are distributed evenly within the three slope units, while in the agroforestry area, the processes show uneven distribution. In the cassava field, the residual zone is dominated by rill erosion and cracks, the erosional zone is dominated by gully erosion, cracks, landslide reactivation, and material displacement, and the depositional zone is dominated by gully erosion, landslide reactivation, and soil creep. On the other hand, in the agroforestry area, rill erosion and landslide reactivation lie along the erosional and depositional zone, while gully erosion and landslide reactivation are scattered on the depositional zone.

In other areas of Java, erosional and depositional zones in the former landslide area were often found to be repeated due to more than one landslide occurrence [31]. Along with the number of the slope unit repetitions, the erosional zone becomes shorter, while the depositional zone becomes longer. On a long slope, the lowest depositional zone slope has a slope angle close to zero, which is located at the bottom of the valley and often associated with the water bodies.

In general, landslides in a very thick soil will form a rotational landslide type that moves large amounts of soil material from the upper slope to the lower slope, then generates typical slope units. The part of the landslide with removed surface material is defined as an erosional zone, while the part with buried original soil material is defined as a depositional zone. On the other hand, the part of the landslide with thick soils, that has not been disturbed by the landslide process, is defined as a residual zone. Among all the three zones, the erosional zone has the largest slope angle, then followed by depositional and residual zones. Besides, in terms of area coverage, the residual zone commonly

has the smallest relative area compared to other zones (Table 1). The zone classification or the landslide morphological order was based on the order III landscape classification by [34], which is not specifically discussed in recent studies [35–37].

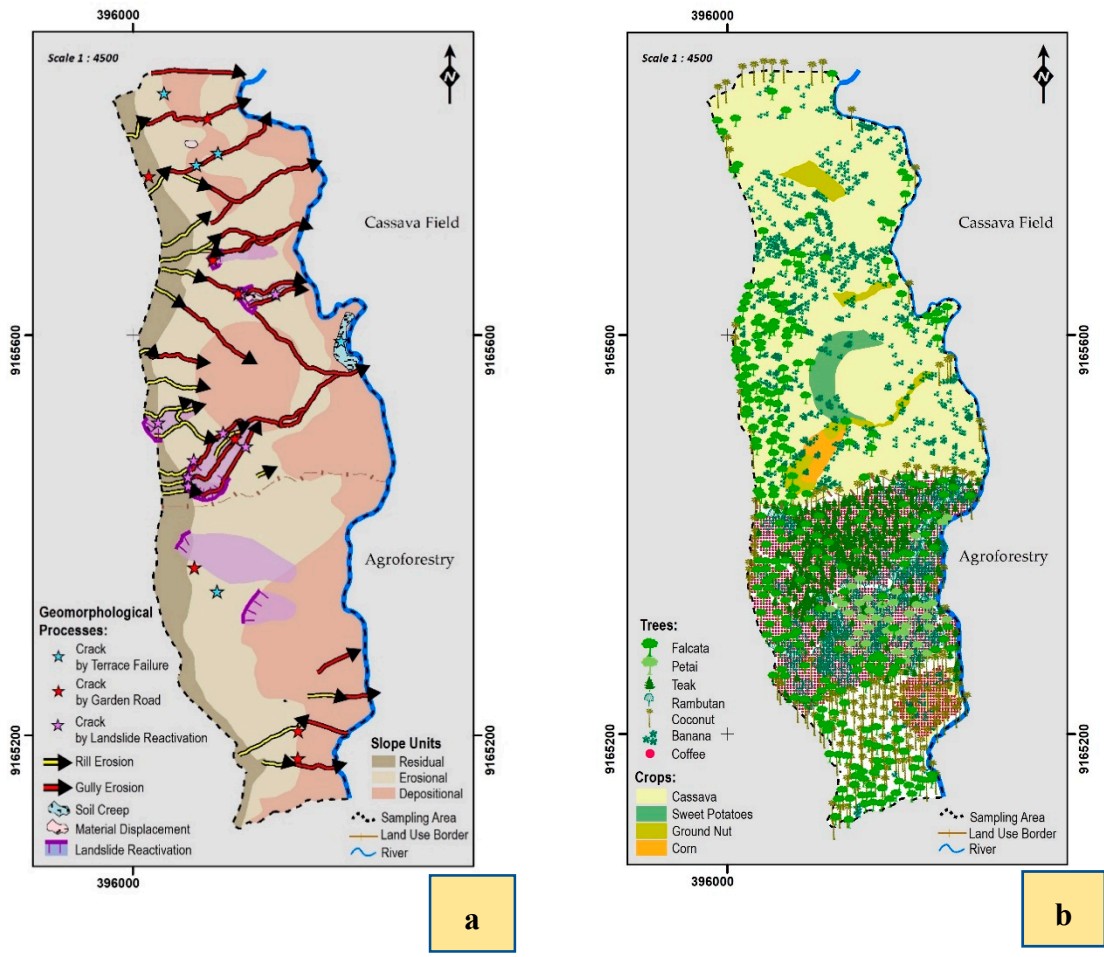

**Figure 3.** (**a**) Geomorphological process results within each slope zone, (**b**) Trees and crops distribution in the sampling area.

**Table 1.** Zones of the Slope Units that Formerly Experienced Landslide.

| Land Use | Slope Zones | Relative Position at the Landslide | Slope Length (m) | Slope Width (m) | % Area Coverage |
|---|---|---|---|---|---|
| | Residual | Upper | 38 | 352 | 12.37 |
| Agroforestry | Erosional | Middle | 156 | 359 | 45.91 |
| | Depositional | Lower | 124 | 346 | 41.70 |
| | Residual | Upper | 32 | 432 | 12.28 |
| Cassava Field | Erosional | Middle | 120 | 443 | 47.08 |
| | Depositional | Lower | 48 | 442 | 40.62 |

### 3.2. Types and Intensity of the Geomorphological Process

According to the types and sizes of the geomorphological processes, which were tabulated based on land use and slope units, the typical processes in the cassava field are rill erosion, gully erosion, soil creep, and landslide reactivation, while in the agroforestry area, the processes found are gully erosion and landslide reactivation. In the cassava field, rill erosion is dominantly located in the residual zone, gully erosion and landslide reactivation in the erosional and depositional zone, and soil creep in the depositional zone, whereas in the agroforestry area, the gully erosion and landslide reactivation are found in the depositional zone. Once a landslide occurs, other geomorphological processes that

work will continue until they cause the next massive landslide [20,38,39]. A massive landslide process produces a typical slope arrangement, followed by the decreased intensity of the next landslide reactivation. All the mechanisms indicate that the various types and intensities of geomorphological processes may trigger the landscape changes. The various types of geomorphological processes and the intensity with which they occur in the sample site are presented in Figure 4.

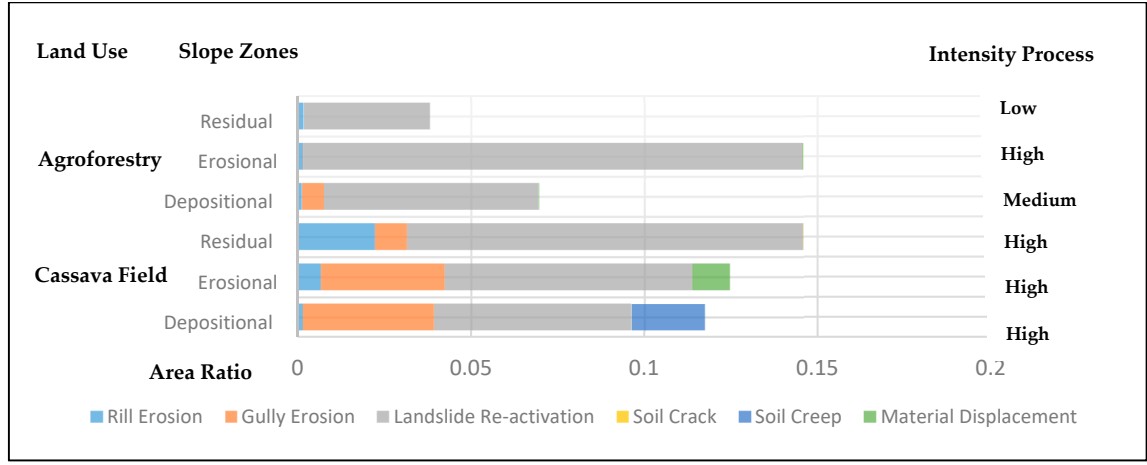

**Figure 4.** The ratio of geomorphological process areas in each land surface morphological zone of agroforestry and cassava field.

In all slope units of the cassava field, the material movement from one slope unit to a lower slope unit is classified as a high intensity, which is slightly different from the process intensity in agroforestry land use. The intensity difference occurs because of yearly disturbance to the cassava field as a result of cultivated cassava rotation, while the disturbance to agroforestry land does not occur every year. The forms of disturbance include soil tillage, material addition, changes in the area of vegetation cover, and a reduction in slope length and angle due to terraces' development affecting surface runoff concentration. Therefore, it can be concluded that the material removal in the cassava field is a result of loose soil characteristics, continuously changed vegetation cover, and the presence of concentrated surface runoff in certain furrows. This is reinforced by dry and rainy season observations that indicated that erosion channel arrangement can be changed due to yearly soil cultivation, especially for those starting at the beginning of the rainy season. In contrast, the material transfer in the agroforestry land is only high in the middle slope (erosional zone) that has a large slope angle and length.

The random plant combinations in the agroforestry land were found to be ineffective since the moderate to high geomorphological processes were still present in the erosional and depositional zones. This condition is triggered by a simultaneous geomorphological process from the three slopes' units, thus the accumulative effect occurred on the lower part of erosional zones and depositional zones. The moderate geomorphological process on erosional zone tends to happen due to land management, which modifies water flow direction to be straight, causing flow concentration absence and low erosion. On the contrary, an intensive geomorphological process is formed on the depositional zone due to flow concentration as a consequence of slope angle change. This mechanism proves that appropriate management interventions can reduce the geomorphological processes' intensity.

*3.3. Inventory of Recent Vegetation Cover*

The results of the plant species inventory in the sampling area were grouped into perennial plants, that are considered as trees, and seasonal plants, that are considered as crops. The groups were further classified according to their position on slope sections and land use units, as presented in Table 2a,b and as depicted in Figure 3b. The plant classification aimed to describe the effect of plants on geomorphological process forms and types.

**Table 2.** (**a**) Trees in the Sampling Area of Agroforestry and Cassava Field. (**b**) Crops in the Sampling Area of Agroforestry and Cassava Field.

(a)

| Land Use | Slope Zones | ∑Falcata | ∑Coconut | ∑Petai | ∑Teak | ∑Rambutan | ∑Bamboo | ∑Banana | ∑Coffee | Total Number of Trees | Trees Density | Canopy Cover Per Zone Area (%) |
|---|---|---|---|---|---|---|---|---|---|---|---|---|
| Agroforestry | Residual | 38 | 23 | 0 | 13 | 10 | 0 | 0 | 486 | 570 | 0.0844 | 79 |
| | Erosional | 67 | 64 | 5 | 238 | 69 | 0 | 7 | 1952 | 2402 | 0.0959 | 84 |
| | Depositional | 55 | 78 | 33 | 41 | 104 | 7 | 41 | 1666 | 2025 | 0.0890 | 80 |
| Sub-Total | | **160** | **165** | **38** | **292** | **183** | **7** | **48** | **4104** | **4997** | **0.0916** | **81** |
| Cassava Field | Residual | 40 | 16 | 0 | 0 | 0 | 0 | 92 | 0 | 148 | 0.0157 | 28 |
| | Erosional | 53 | 5 | 0 | 0 | 0 | 0 | 360 | 0 | 418 | 0.0116 | 11 |
| | Depositional | 22 | 11 | 0 | 0 | 0 | 5 | 257 | 0 | 295 | 0.0095 | 10 |
| Sub-Total | | **115** | **32** | **0** | **0** | **0** | **5** | **709** | **0** | **861** | **0.0113** | **13** |

(b)

| Land Use | Slope Zones | ∑Cassava | ∑Sweet Potatoes | ∑Corn | ∑Ground Nut | Total Number of Crops | Crops Density | Canopy Cover Per Zone Area (%) |
|---|---|---|---|---|---|---|---|---|
| Agroforestry | Residual | 0 | 0 | 0 | 0 | 0 | 0.0000 | 0 |
| | Erosional | 24 | 0 | 0 | 0 | 24 | 0.0010 | 0 |
| | Depositional | 3040 | 0 | 0 | 0 | 3040 | 0.1337 | 7 |
| Sub-Total | | **3064** | **0** | **0** | **0** | **3064** | **0.0562** | **3** |
| Cassava Field | Residual | 8889 | 0 | 0 | 125 | 9014 | 0.9590 | 47 |
| | Erosional | 33,942 | 0 | 840 | 306 | 35,088 | 0.9745 | 59 |
| | Depositional | 26,452 | 11,172 | 314 | 392 | 38,330 | 1.2335 | 66 |
| Sub-Total | | **69,283** | **11,172** | **1154** | **823** | **82,432** | **1.0778** | **60** |

All types of plants in the sampling area have different economic, social and ecological values, both in the agroforestry land and the cassava field. Trees and crops cultivated in the agroforestry land provide economic value that is mainly used for daily and seasonally consumption, while cassava plantation is mainly for annual income to cover high household expenses, such as the school fees of the children. In terms of social values, the agroforestry production provides continuous economic value, supporting many traditional events such as a funeral, wedding, and other ceremonies [40], while the cassava, as a local food product, supports food security systems both in the study area and the surrounding area. Besides, in terms of ecological function, the agroforestry plays a more important role in controlling the soil erosion than the cassava plantation does, due to its relatively intensive monoculture systems. Thus, some conservation techniques, such as gully plug, using tree barriers and terrace management, should be applied in the cassava field to conserve soil resources. Besides, the variety of plants cultivated on privately owned agroforestry land also have ecological value due to the combination of trees and crops.

## 4. Discussion

The existing plant distribution in cassava field and agroforestry is presented in Figure 3a,b. Cassava plants are planted randomly in all slope units of cassava field. The actual plant layout in the agroforestry land use can also be seen in Figure 5. The Falcata plants are randomly distributed on the residual and erosional slope units in the agroforestry. The multilayer trees sequentially consist of Falcata–Petai–Teak–Rambutan and Coffee plants. The presence of Falcata plants affects slope stability, since the Falcata roots have reduced root density in soil depths of 30–40 cm [41]. The logging pattern of Falcata wood in every 5–10 years at the research location has left roots in the soil that ease and decompose after 2–3 years. Those conditions cause coarse soil pores that increase the rainwater infiltration process and makes the subsoil over-wet. The lower soil layer in the study site consists of high clay content [42] that creates a landslide slip plane when it is wet. The selection of plant species needs to be re-arranged to create an ideal balance between geomorphological control processes and land cultivation results in an agroforestry system. The actual ratio area of rill erosion, gully erosion, landslide reactivation and soil creep between cassava field and agroforestry might become a tool for the effectiveness of the proposed plant arrangement.

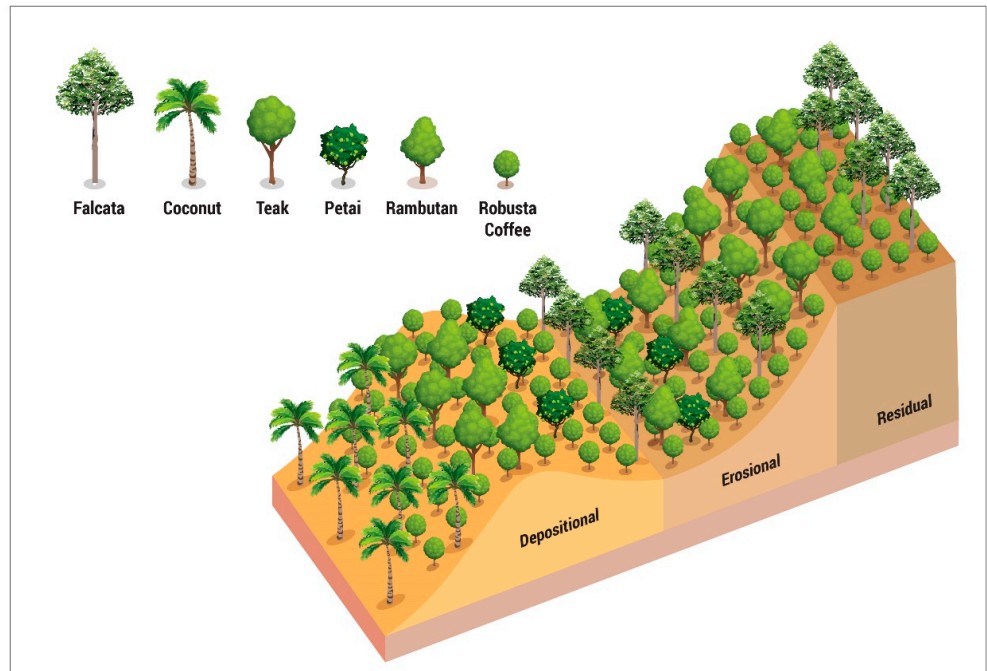

**Figure 5.** Actual plant layout in the existing agroforestry.

Plant species inventory in the sampling area and its surroundings was intended to determine the recommended plants based on the land quality. According to the inventory results, 37 types of plant were cultivated randomly by the community without considering the slope layout and the existing geomorphological processes' sequence. The study grouped those existing plants according to plant morphology, ecological functions and economic functions, to form a base recommendation of vegetation arrangement on each slope unit (Table 3). The plant species selection for each slope unit aimed to provide an optimum ecological function controlling the geomorphological processes' intensity [43]. The layout design of plant arrangement in the existing agroforestry is shown in Figure 5.

The residual zones need to be conserved using plants with strong roots and thick leaves, tha minimize the impact of soil detachment and transport caused by splash, sheet, and rill erosion. Besides, cultivating dry-resistant plants also needs to be considered due to the lack of water during the dry season in the residual zones. The recommended plants are Indonesian rosewood as tall plants, coffee trees as intercrop plants with a medium height, and herb plants as root crops. A combination of those plants will give an optimum ecological function in protecting the soil surface from rainwater erosion [44,45]. In addition to an optimum ecological function, those three types of plant also provide different economic functions in different periods of time. Indonesian rosewood plants can be harvested within ten years, coffee trees within a year, and herb plants within three years

The erosional zones need to be conserved with plants that minimize the impact of soil structure downslope movement caused by rill erosion, gully erosion, reactivation of landslides, and material displacement. The plant species to consider is the plant that can increase the slope stability with its roots' ability to penetrate the soil layer both vertically and laterally [46]. The species can be groups of trees that produce wood as a function of the decades' production and/or groups of trees that produce fruit as a function of annual or seasonal production. The options for wood plants include teak [47,48], mahogany, and Indonesian rosewood, while the fruit crops include mangosteen, dog fruit, stink bean, jackfruit, lanzones/duku, and durian [49]. Besides, the intercrops that can be cultivated are coffee as a medium height plant [50–52] and herbs as root crops [53].

A depositional zone is a slope unit that has a small slope angle with a very thick loose material so that it has a high infiltration capacity. The loose material originates from various geomorphological processes that work in the erosional zone and the residual zone [54]. Due to the extremely high soil water content, the depositional zones need to be controlled with plants that minimize the intensity of gully erosion, soil cracks, landslides, and soil creeps [55]. The high water content can be maintained by cultivating high water consumption plants such as bamboo [56–58], banana [59,60], sugar palm [61,62], and coconut [63]. They can be planted at the lowest elevation of the depositional zone and can be combined with the recommended crop choices for erosional zones. Besides, the presence of trees and plants in the depositional zone may improve infiltration capacity [64]. Therefore, various other lowland plants such as grasses, that function as livestock forages, can also be planted as intercrops. The recommended plant layout, with economic, social, and ecological functions, is presented in Figure 6.

**Table 3.** Recommendation of the Selection and Arrangement of Local Vegetation Based on the Slope Unit Zones.

| No | Slope Zone Location | Layer Class Position | Plants Name | Plant's Morphology | | | | | | | | Ecological Function | | | Economy Function | |
|----|----|----|----|----|----|----|----|----|----|----|----|----|----|----|----|----|
| | | | | Height (m) | Canopy Shape | Stem Diameter (cm) | Stem Straightness | Root | Leaf Shape | Leaf Size (L × W cm) | Soil Conservation | Water Conservation | Environmental Adaptation | Economy Value | Harvest Age |
| 1 | R/E/D | A | Galangal (*Kaempferia galanga*) | 0.05 | Bush | - | Clumps | Rhizome | Oval | 10 × 5 | Ground base cover crop | Reducing runoff velocity | Shade-resistant | Seasonal crop | 6–12 months |
| 2 | R/E/D | A | Ginger Zerumbet (*Zingiber zerumbet*) | 1 | Bush | 5 | Clumps | Rhizome | Oval | 30 × 5 | Ground base cover crop | Reducing runoff velocity | Shade-resistant | Seasonal crop | 10 months |
| 3 | R/E/D | A | Curcuma (*Curcuma zanthorrhiza*) | 1 | Bush | 5 | Clumps | Rhizome | Oval | 50 × 20 | Ground base cover crop | Reducing runoff velocity | Shade-resistant | Seasonal crop | 9–10 months |
| 4 | R/E/D | A | Turmeric (*Curcuma longa*) | 1 | Bush | 5 | Clumps | Rhizome | Oval | 50 × 20 | Ground base cover crop | Reducing runoff velocity | Shade-resistant | Seasonal crop | 11–12 months |
| 5 | R/E/D | A | Galangal (*Alpinia galanga*) | 1 | Bush | 5 | Clumps | Rhizome | Oval | 40 × 10 | Ground base cover crop | Reducing runoff velocity | Shade-resistant | Seasonal crop | 10–12 months |
| 6 | R/E/D | A | Ginger (*Zingiber officinale*) | 1 | Bush | 5 | Clumps | Rhizome | Pinnate | 15 × 10 | Ground base cover crop | Reducing runoff velocity | Shade-resistant | Seasonal crop | 8 months |
| 7 | R/E/D | B | Robusta Coffee (*Coffea robusta*) | 5 | Oval | 5–10 | Shrubs | Tap roots | Oval | 20 × 5 | Medium ground cover crop | Reducing runoff velocity | | Perennial crop | 3 months |
| 8 | R | A | Pineapple (*Ananas comusus*) | 1 | Bush | - | Clumps | Fibrous roots | Pinnate | 150 × 7 | Ground base cover crop | Reducing runoff velocity | | Seasonal crop | 24 months |
| 9 | R | C | Gamal (*Gliricidia sepium*) | 6 | Oval | 5–30 | Branchy | Tap roots | Pinnate | 6 × 10 | Wind breaker, Medium ground cover crop | | Dry-resistant plant | Perennial crop | 3–4 months |
| 10 | R | C | Water Apple (*Syzygium aqueum* | 10 | Dome | 50–70 | Branchy | Tap roots | Oval | 10 × 6 | Medium ground cover crop | | | Perennial crop | 3–4 years |
| 11 | R | C | Avocado (*Persea americana*) | 10 | Dome | 50–70 | Branchy | Tap roots | Oval | 15 × 5 | Medium ground cover crop | | | Perennial crop | 5–10 years |
| 12 | R | C | Mango (*Mangifera indica*) | 10 | Dome | 50–70 | Branchy | Tap roots | Oval | 20 × 10 | Medium ground cover crop | | | Perennial crop | 3–4 years |

**Table 3.** *Cont.*

| No | Slope Zone Location | Layer Class Position | Plants Name | Plant's Morphology | | | | | | | | Ecological Function | | | Economy Function | |
|---|---|---|---|---|---|---|---|---|---|---|---|---|---|---|---|---|
| | | | | Height (m) | Canopy Shape | Stem Diameter (cm) | Stem Straightness | Root | Leaf Shape | Leaf Size (L × W cm) | | Soil Conservation | Water Conservation | Environmental Adaptation | Economy Value | Harvest Age |
| 13 | R | C | Rambutan (*Nephelium lappaceum*) | 14 | Dome | 25–30 | Branchy | Tap roots | Elliptic | 10 × 5 | | Medium ground cover crop | | Adaptable | Perennial crop | 6–7 years |
| 14 | E | C | Langsat (*Lansium parasiticum*) | 15 | Oval | 50–70 | Branchy | Tap roots | Pinnate compound | 10 × 5 | | High ground cover crop | | | Perennial crop | 8–12 years |
| 15 | E | C | Mangosteen (*Garcinia mangostana*) | 15 | Dome | 50–70 | | Tap roots | Elliptic | | | High ground cover crop | | | Perennial crop | 5 years |
| 16 | E | D | Jengkol (*Archidendron pauciflorum*) | 20 | Dome | | Straight | Tap roots | Compound | 15 × 10 | | High ground cover crop | | | Perennial crop | 4 years |
| 17 | E | D | Gnetum (*Gnetum gnemon*) | 20 | Oval | 10–20 | Straight | Tap roots | Elliptic | 10 × 5 | | High ground cover crop | | Dry plants | Perennial crop | 5–7 years |
| 18 | E | D | Jackfruit (*Artocarpus heterophyllus*) | 20 | Dome | 30–100 | Straight | Tap roots | Elliptic | 15 × 5 | | High ground cover crop | | | Perennial crop | 5–10 years |
| 19 | E | D | Teak (*Tectona Grandis*) | 20 | Oval | 50–150 | Straight | Tap roots | Rounded | 30 × 20 | | High ground cover crop | Deciduous during dry season | Dry plants | Perennial crop | 20–80 years |
| 20 | E | D | Duku (*Lancium domesticum*) | 20 | Oval | | Branchy | Tap roots | Pinnate compound | 15 × 5 | | High ground cover crop | | Shade-resistant | Perennial crop | 15 years |
| 21 | E | D | Mahogany (*Swietenia mahagoni*) | 20 | Dome | 120 | Straight | Tap roots | Rounded compound | 8 × 4 | | High ground cover crop | | | Perennial crop | 10 years |
| 22 | E | E | Durian (*Durio zibethinus*) | 35 | Dome | 70–100 | Branchy | Tap roots | Oval | 20 × 10 | | High ground cover crop | | Good planted in thick soil | Perennial crop | 8–15 years |
| 23 | D | A | Ground Nut (*Arachis hypogaea*) | 0.25 | Bush | 1 | Clumps | Tap Roots | Compound | 5 × 2 | | Ground base cover crop, nitrogen fixing | Reducing runoff velocity | | Seasonal crop | 3 months |
| 24 | D | A | Sweet Potatoes (*Ipomoea batatas*) | 0.3 | Bush | 1 | Clumps | Tuber | Fig leaf | 4 × 3 | | Ground base cover crop | Reducing runoff velocity | | Seasonal crop | 3–4 months |

**Table 3.** *Cont.*

| No | Slope Zone Location | Layer Class Position | Plants Name | Plant's Morphology | | | | | | | | Ecological Function | | | Economy Function | |
|---|---|---|---|---|---|---|---|---|---|---|---|---|---|---|---|---|
| | | | | Height (m) | Canopy Shape | Stem Diameter (cm) | Stem Straightness | Root | Leaf Shape | Leaf Size (L × W cm) | Soil Conservation | Water Conservation | Environmental Adaptation | Economy Value | Harvest Age |
| 25 | D | A | Grass (*Panicum muticum*) | 1 | Bush | 3 | Clumps | Fibrous roots | Pinnate | 100 × 5 | Ground base cover crop | Reducing runoff velocity | Wet plants | Seasonal crop | 25–30 days |
| 26 | D | B | Corn (*Zea mays*) | 2 | Shrub | 5 | Straight | Fibrous roots | Pinnate | 30 × 10 | Ground base cover crop | | | Seasonal crop | 3 months |
| 27 | D | B | Banana (*Musa spp.*) | 2 | Rounded | 30 | Straight | Fibrous roots | Long oval | 120 × 50 | Medium ground cover crop, Soil moisture balancer | Soil water extraction | | Seasonal crop | 12 months |
| 28 | D | B | Cassava (*Manihot utilissima*) | 2 | Shrub | 3 | Straight | Tuber | Fig leaf | 20 × 10 | Medium ground cover crop | | Dry plants | Seasonal crop | 9–10 months |
| 29 | D | B | Papaya (*Carica papaya*) | 5 | Rounded | 30–40 | Straight | Fibrous roots | Fig leaf | 50 × 70 | Medium ground cover crop | | Wet plants | Seasonal crop | 8–9 months |
| 30 | D | C | Breadfruit (*Artocarpus altillis*) | 13 | Dome | 50–70 | Branchy | Tap roots | Pinnate | 40 × 30 | Medium ground cover crop | | | Perennial crop | 4 years |
| 31 | D | C | Petai (*Parkia speciosa*) | 15 | Dome | 20 | Branchy | Tap roots | Pinnate compound | 0.1 × 0.05 | High ground cover crop | | Wet plants | Perennial crop | 4–10 years |
| 32 | D | D | Acacia (*Acacia auriculiformis*) | 17 | Dome | 10–20 | Straight | Tap roots | Pinnate compound | 10 × 2 | High ground cover crop | | | Perennial crop | 5 years |
| 33 | D | D | Bamboo (*Dendrocalamus asper*) | 20 | Oval | 12–20 | Straight | Fibrous roots | Pinnate | 25 × 5 | Holding soil, protecting riverbanks, preventing landslides, controlling erosion, Soil moisture balancer | Soil water extraction | | Perennial crop | 3–6 years |
| 34 | D | D | Sugar Palm (*Arenga pinnata*) | 25 | Palm | 65 | Straight | Fibrous roots | Pinnate | 8 × 5 | Holding soil, riverbanks preventing landslides, controlling erosion, Soil moisture balancer | Soil water extraction | Low evapotranspiration | Perennial crop | 6–12 years |

**Table 3.** *Cont.*

| No | Slope Zone Location | Layer Class Position | Plants Name | Plant's Morphology | | | | | | | | | | | Economy Function | |
|---|---|---|---|---|---|---|---|---|---|---|---|---|---|---|---|---|
| | | | | Height (m) | Canopy Shape | Stem Diameter (cm) | Stem Straightness | Root | Leaf Shape | Leaf Size (L × W cm) | Soil Conservation | Water Conservation | Environmental Adaptation | | Economy Value | Harvest Age |
| 35 | D | E | Coconut (*Cocos nucifera*) | 30 | Palm | 30–40 | Straight | Fibrous roots | Pinnate | 10 × 5 | Holding soil, Soil moisture balancer | Soil water extraction | Good planted on flat land | | Perennial crop | 5–10 years |
| 36 | R/D | E | Rosewood (*Dalbergia latifolia*) | 30 | Dome | 70–100 | Straight | Tap roots | Pinnate | 10 × 5 | High ground cover crop, Nitrogen fixing | | Need good drainage | | Perennial crop | 20–50 years |
| 37 | D | E | Falcata (*Albizia falcataria*) | 35 | Dome | 70–100 | Straight | Tap roots | Pinnate compound | 0.1 × 0.05 | High ground cover crop, Nitrogen fixing | | | | Perennial crop | 7 years |

Layer Class Position: Layer A = 0–1 m (Ground Base); Layer B = 1–5 m (First Layer); Layer C = 5–15 m (Second Layer); Layer D = 15–25 m (Third Layer); Layer E = 25–35 m (Fourth Layer).
Slope Zones: R = Residual Zone; E = Erosional Zone; D = Depositional Zone.

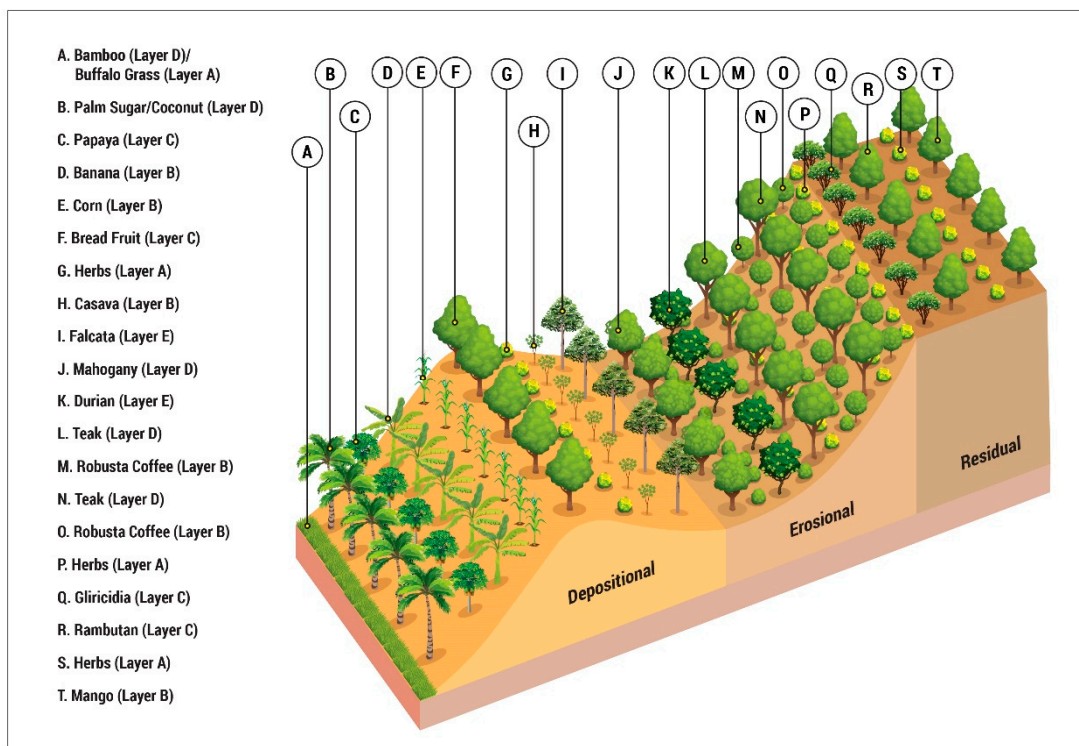

**Figure 6.** Illustration of Plant Layout Model of Agroforestry Systems to Control Landslides following three succession slope units of landslide.

The vegetation arrangement, cultivated on slope units of a former landslide, has an ecological function to reduce the threat of landslides. It provides lateral protection from the residual zone to the depositional zone and also provides vertical protection on the soil surface and the subsoil. The trees and crops work together to protect the soil from direct exposure to raindrops so the rate of erosion can be reduced. Besides, plants with different levels in height also have different root depths, thus the soil will not be easily saturated, which can trigger a landslide. Therefore, the land is relatively undisturbed by adverse geomorphological processes so that plants can have optimum productivity. As a result, the optimum plant productivity results in many different products that guarantee a sustainable economic function from season to season and year to year. On the other hand, the function of plants is not only limited to economic reasons but also needed for social security. The sustainable economic function ensures social security in relation to certain events that are held, as even the executants do not own sufficient money in their daily life. It maintains a subsistent source of income in a community and affects the sustainability of a better social life because the traditional community prefers to save things rather than money [65].

## 5. Conclusions

Landslide reactivation consists of processes, such as rill erosion, gully erosion, and soil creeps, that are sequentially located along the residual zone, erosional zone, and depositional zone. The simultaneous vegetative technique at the former landslide area was considered as an effective technique to control erosion processes, thus landslide reactivation could be managed. In order to minimize landslide reactivation threat, the vegetative technique should prioritize trees' and crops' ecological functions to reduce runoff, water absorption, and soil moisture absorption. Besides considering the ecological functions, the study also took into account the selection of trees and crops based on the importance of economic and social values for the community.

**Author Contributions:** Data collection, analysis, visualization, writing—original draft preparation was done by R.P.; formulating and conceptualization of methodology as well as writing the final draft, review and editing, small grand funding acquisition were mainly done by J.S.; field work data collection and supervision were mainly done by M.A.S.; All authors have read and agreed to the published version of the manuscript.

**Funding:** The research was self-financed by all three involved researchers. The second and the third researcher have a personal budget to support the research both in the field and laboratory. However, the manuscript preparation and publication may get partial support from Universitas Gadjah Mada whenever it is accepted for publication.

**Acknowledgments:** The researcher would like to express great appreciation to all members of the Transbulent (Transition of Natural Processes in the Built-up Environment) research group for their nice discussion and for providing support during the field survey.

**Conflicts of Interest:** The authors declare no conflict of interest.

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
