# Peer review of "Trees and Crops Arrangement in the Agroforestry System Based on Slope Units to Control Landslide Reactivation on Volcanic Foot Slopes in Java, Indonesia"

_land, doi:10.3390/land9090327_

Round 1

Reviewer 1 Report

Manuscript ID: land-898522

Title: Trees and Crops arrangement in the agroforestry system based on slope units to control landslide reactivation on volcanic foot slopes in Java, Indonesia

Research is interesting and timely, referring to trees and crops arrangement in the agroforestry system based on slope units to control landslide reactivation on volcanic foot slopes in Java, Indonesia. The major limitations consist in having considered landslides in general without differentiating their typology and, therefore, their different spatial expression. I suggest integrating the material used by an appropriate geomorphological mapping and use this to differentiate landslide for typology and activity state. In the following, more specific comments.

1- The proposed approach should be compared with several state-of-the-art approaches, so we can evaluate the novelty of the proposed approach. 

2- In the experiments, it is interesting to see the result of aerial photo interpretation.

Abstract:

  • You must Summarize the article's main findings and Indicate the main conclusions

Introduction:

  • Lines 42-49: each sentence starts with “agricultural sector development”
  • Lines 53-54: the references not in a proper style
  • Figure 1: the map needs a scale bar; coordinate system; north arrow; as well as the text not clear for readability.

Materials and Methods:

  • Line 75-82: this paragraph fits with the introduction section not here.
  • Figure 2: the text needs to be enlarged for more clarity, scale, and legend reflect the color variation (elevation)

Results:

  • Line 135, remove the space  
  • Lines 150-151: it is not clear what do you mean by order III landscape classification.
  • Line 195: “Non-Permanent Trees in”

Discussion:

  • Figure 4: the text is not read easily on the figure.

Author Response

Research is interesting and timely, referring to trees and crops arrangement in the agroforestry system based on slope units to control landslide reactivation on volcanic foot slopes in Java, Indonesia. The major limitations consist in having considered landslides in general without differentiating their typology and, therefore, their different spatial expression. I suggest integrating the material used by an appropriate geomorphological mapping and use this to differentiate landslide for typology and activity state. In the following, more specific comments.

1- The proposed approach should be compared with several state-of-the-art approaches, so we can evaluate the novelty of the proposed approach. 

Response: The additional proposed approach is outlined in the line of 63 – 70.

2- In the experiments, it is interesting to see the result of aerial photo interpretation.

Response: Additional text about the interpretation result available in line of 135 – 143 for geomorphological processes. We also display the maps from aerial photo interpretation (Figure 3a and 3b)

Abstract:

  • You must Summarize the article's main findings and Indicate the main conclusions

Response: It has been mentioned in the line of 24 - 28

Introduction:

  • Lines 42-49: each sentence starts with “agricultural sector development”

Response: Replaced with “The agricultural production escalation” (line 46)

  • Lines 53-54: the references not in a proper style

Response: It has been replaced with proper style (line 52)

  • Figure 1: the map needs a scale bar; coordinate system; north arrow; as well as the text not clear for readability.

Response: The map has been replaced with additional legend and highest resolution

Materials and Methods:

  • Line 75-82: this paragraph fits with the introduction section not here.

It has been replaced in the Introduction, last paragraph (line of 80 - 85)

  • Figure 2: the text needs to be enlarged for more clarity, scale, and legend reflect the color variation (elevation)

The map has been replaced with additional legend, edited texts and highest resolution

Results:

  • Line 135, remove the space  

It has been removed

  • Lines 150-151: it is not clear what do you mean by order III landscape classification.

This is already mentioned in the reference number 32 (Lobeck 1939). The 3rd order of landscape is detail landscape as a result of exogenic processes (line of 160-162).

  • Line 195: “Non-Permanent Trees in”

We already add “in” and use only Trees (as suggested from other reviewer), line of 210 and 213

Discussion:

  • Figure 4: the text is not read easily on the figure.

We replaced the figure with little modification and highest resolution (the number of figure is changed to Figure 5/line 278)

Reviewer 2 Report

Please highlight what are main scientific contributions of this work.

What is the main scientific novelty in this paper? Please explain and reflect that in the manuscript.

Author Response

Please highlight what are main scientific contributions of this work.

Response:

The highlight is mentioned in the abstract (line of 15 – 17)

What is the main scientific novelty in this paper? Please explain and reflect that in the manuscript.

Response:

The scientific novelty is mentioned in the abstract (line of 23-24)

Reviewer 3 Report

Purwaningsih et al. present a manuscript evaluating different planting schemes of agroforestry systems in preserving the landscape in Java from erosion and even catastrophic landslides. A large range of crops and tree species is cover in their inventory. Management recommendations are made in the discussion and in limited extend in the conclusion section (the conclusions sounds a little bit repetitive to the abstract – this should be improved) .

The manuscript is generally well written and the findings are nicely presented. Maybe the figure should be provided in higher quality in particular the legend of figure 4 is really hard to read – but it is a very crucial figure. Can’t the authors add a description like in  figure 2.

Author Response

Purwaningsih et al. present a manuscript evaluating different planting schemes of agroforestry systems in preserving the landscape in Java from erosion and even catastrophic landslides. A large range of crops and tree species is cover in their inventory. Management recommendations are made in the discussion and in limited extend in the conclusion section (the conclusions sounds a little bit repetitive to the abstract – this should be improved) .

Response:

The conclusion has been improved (line of 296 – 303)

The manuscript is generally well written and the findings are nicely presented. Maybe the figure should be provided in higher quality in particular the legend of figure 4 is really hard to read – but it is a very crucial figure. Can’t the authors add a description like in  figure 2.

Response:

For figure 5, I replaced the figure with little modification and highest resolution.

Figure 5 additional description: line of 278 - 279

Reviewer 4 Report

Current work attempts a joint analysis of aerial image analysis and field work on plant record in a volcanic study site. Although it has the potential to evolve into an interesting work, it encounters significant issues which confine this potential.

Primarily the text needs extensive revision in terms of language. Is several cases the communicated idea is difficult to be understood due to poor quality of the language.   

Structural revisions of the sections is also needed. For example in the Abstract the aim of the research, the research question and the implications/ scientific contribution of the work are not clearly stated. 

In the Introduction some important background information which is provided later on in the text (i.e. the two types of land tenure - private and governmental or the types of rain-fed agriculture - although spotted in figure 1 nothing is mentioned in the text) is missing.  Also text improvements can be made (i.e. summarize and write more some repetitions (i.e. l.50 - 58 can be written in a more compact/comprehensive way).

The Materials and methods section has significant deficiencies. 1. Study site description: lacks significant details about the sampling area (size, climatic conditions, soil and geomorphological characteristics of the specific site are not clearly provided - although general description of the area is given) and explanation about why this sampling area can represent the broader area. 2. Terminological issues: scientific terminology and consistency of the terms used. For example the authors use the term "sequential slope units", which could be possibly replaced by the scientific term "Catena". The terms permanent /non-permanent trees/crops should be better specified. Do the authors with the non-permanent refer to annual cultivations? The categorization between the three plant types is not clear (especially between types two and three).   3.Methodological issues: the two approaches (field study and image analysis) are poorly described. Which methods where followed? which tools were used for both image analysis and field studies? concerning the photographs, which were their technical characteristics?

The Results section requires significant restructuring. Despite the interesting Tables and Figures, the text poorly supports them. For example i.142 - 151, 153 - 156 instead of analysing the results provide theoretical information. How does the information presented in Figure three derives?

In l.166 is mentioned "cultivated cassava rotation". Does this mean that cassava belongs to a crop rotation system? What other plants are used in this system?

In l.168 is mentioned "reduction of slope length and angle". How was this measured?

l.175 - 178 seem to have a repetition.

l.180 - 181: Which are these modifications? Is there some study which shows some results from the impact of these measures in the area? 

Tables 2a and 2b. be careful when using the term "tree". i.e. potatoes and corns are not trees. Also how can a tree be considered as a non-permanent (?annual) plant specie?

l.197 - 207 need rephrasing for clearer meaning.

In the Discussion section is mainly discussed the ecological aspect of the analysis, although in the Abstract and Introduction it is reported that the study will look the combined ecological and socioeconomic framework of the study site.  How is the sentence of l. 271 - 274 supported by current analysis?

Conclusions: how is the statement provided in l.279 - 280 supported by current research? Is there some other research supporting this statement?

In l. 282 is mentioned "social functions". Which area these? Please clarify.

Author Response

Current work attempts a joint analysis of aerial image analysis and field work on plant record in a volcanic study site. Although it has the potential to evolve into an interesting work, it encounters significant issues which confine this potential.

Primarily the text needs extensive revision in terms of language. Is several cases the communicated idea is difficult to be understood due to poor quality of the language.   

Structural revisions of the sections is also needed. For example in the Abstract the aim of the research, the research question and the implications/ scientific contribution of the work are not clearly stated. 

The abstract has been improved

In the Introduction some important background information which is provided later on in the text (i.e. the two types of land tenure - private and governmental or the types of rain-fed agriculture - although spotted in figure 1 nothing is mentioned in the text) is missing. 

To avoid misunderstanding, we prefer to not use private and governmental type of land tenure.

Figure 1 is improved with additional legend and text improvement.

Also text improvements can be made (i.e. summarize and write more some repetitions (i.e. l.50 - 58 can be written in a more compact/comprehensive way).

It has been re-write (line of 49-57)

The Materials and methods section has significant deficiencies. 1. Study site description: lacks significant details about the sampling area (size, climatic conditions, soil and geomorphological characteristics of the specific site are not clearly provided - although general description of the area is given) and explanation about why this sampling area can represent the broader area.

This section has been described in the cover letter

2. Terminological issues: scientific terminology and consistency of the terms used. For example the authors use the term "sequential slope units", which could be possibly replaced by the scientific term "Catena".

This section has been described in the cover letter

The terms permanent /non-permanent trees/crops should be better specified. Do the authors with the non-permanent refer to annual cultivations? The categorization between the three plant types is not clear (especially between types two and three).   

The non-permanent trees are referring to seasonal cultivations. To avoid some misunderstanding about plant types, we decide to use trees and crops

Table 2a: Trees

Table 2b: Crops

3.Methodological issues: the two approaches (field study and image analysis) are poorly described. Which methods where followed? which tools were used for both image analysis and field studies? concerning the photographs, which were their technical characteristics?

This section has been described in the cover letter

We are not writing detail about geo-informatics. Related to the material and techniques have been presented briefly (line of 115-132)

The Results section requires significant restructuring. Despite the interesting Tables and Figures, the text poorly supports them. For example i.142 - 151, 153 - 156 instead of analysing the results provide theoretical information. How does the information presented in Figure three derives?

This section has been elaborated in the line of 135 - 143

In l.166 is mentioned "cultivated cassava rotation". Does this mean that cassava belongs to a crop rotation system? What other plants are used in this system?

This section has been described in the cover letter

In l.168 is mentioned "reduction of slope length and angle". How was this measured?

The measurement of reduction of slope length and angle is explained in the line of 187 - 191

l.175 - 178 seem to have a repetition.

It has been modified in line of 198

l.180 - 181: Which are these modifications? Is there some study which shows some results from the impact of these measures in the area? 

This update is associated with previous comment about slope length and angle (line of 187-191 and line 198-203)

Tables 2a and 2b. be careful when using the term "tree". i.e. potatoes and corns are not trees. Also how can a tree be considered as a non-permanent (?annual) plant specie?

The use of “trees” in the table 2b is replaced with “crops”, so that the bananas are categorized as trees.

l.197 - 207 need rephrasing for clearer meaning.

It has been rephrased (line of 214-226)

In the Discussion section is mainly discussed the ecological aspect of the analysis, although in the Abstract and Introduction it is reported that the study will look the combined ecological and socioeconomic framework of the study site.  How is the sentence of l. 271 - 274 supported by current analysis?

The vegetation arrangement, cultivated on slope units of a formerly experienced landslide, has an ecological function to reduce the threat of landslides

This sentence is already explained in the line of 49 –74

Conclusions: how is the statement provided in l.279 - 280 supported by current research? Is there some other research supporting this statement?

This section has been described in the cover letter

Additional figure: map of geomorphological processes result in the two land use

In l. 282 is mentioned "social functions". Which area these? Please clarify.

The term of social function is outlined in the line of 287 - 294

Round 2

Reviewer 4 Report

Most of the comments were addressed. Some minor pending issues are:

Concerning the Abstract structural issues remain. Please consider previous comment. For example, a aim could be the “evaluation of specific vegetation patterns against soil erosion and landslide reactivation”. Moreover, agroforestry is not specifying the relief and the soil types, but rather "the Agroforestry in Java Island".

Minor issues for the Introduction:

l.46 “escalation” is not needed there, please remove.

L51. Landuse – please correct – 2 words.

L84 – growth ? please correct

The comment about the Methodological section was not addressed.

A brief description is required for a sound analysis. L. 115 – 135 for example do not mention anything about the scale and the type of the aerial photographs. These are important characteristics, especially for the interpretation of the geomorphological processes. Indicative literature is:

Nachtergaele, J., & Poesen, J. (1999). Assessment of soil losses by ephemeral gully erosion using high‐altitude (stereo) aerial photographs. Earth Surface Processes and Landforms: The Journal of the British Geomorphological Research Group24(8), 693-706.

Daba, S., Rieger, W., & Strauss, P. (2003). Assessment of gully erosion in eastern Ethiopia using photogrammetric techniques. Catena50(2-4), 273-291

Other minor issues in the Methodological section are:

L 87. Title font formatting is required

l.98 please correct “takes place” some suggestions could be “…the research sampling site is located”, or “ …sampling took place in”

L 115. “some stages”, some is suggested to be replaced by specific number

Minor improvements can still be made in the Result section.

Figure 3 requires some improvement. The legends are hardly readable, in 3a two spots of purple areas do not correspond to some symbol from the legend, 3b is very dense in some parts and the patterns used do not help readability.

L 206 “landuse”, please correct in land use

L220 “soil erosions”, please correct in soil erosion

L221 “trees barriers”, please correct into tree barriers

Author Response

Most of the comments were addressed. Some minor pending issues are:

Concerning the Abstract structural issues remain. Please consider previous comment. For example, a aim could be the “evaluation of specific vegetation patterns against soil erosion and landslide reactivation”. Moreover, agroforestry is not specifying the relief and the soil types, but rather "the Agroforestry in Java Island". – the aim of research is re-writen with additional explanation in the cover letter

Minor issues for the Introduction:

l.46 “escalation” is not needed there, please remove. – the word of “escalation” has been removed

L51. Landuse – please correct – 2 words. – all of “landuse” have been replaced with “land use” (including in the map)

L84 – growth? please correct – the word of “growth” is replaced with “developed”

The comment about the Methodological section was not addressed.

A brief description is required for a sound analysis. L. 115 – 135 for example do not mention anything about the scale and the type of the aerial photographs. These are important characteristics, especially for the interpretation of the geomorphological processes. Indicative literature is:

Some brief description of geo-informatics method has been added in the line of 115 - 119

Nachtergaele, J., & Poesen, J. (1999). Assessment of soil losses by ephemeral gully erosion using high‐altitude (stereo) aerial photographs. Earth Surface Processes and Landforms: The Journal of the British Geomorphological Research Group24(8), 693-706.

Daba, S., Rieger, W., & Strauss, P. (2003). Assessment of gully erosion in eastern Ethiopia using photogrammetric techniques. Catena50(2-4), 273-291

Other minor issues in the Methodological section are:

L 87. Title font formatting is required – the title has been re-formatted

l.98 please correct “takes place” some suggestions could be “…the research sampling site is located”, or “ …sampling took place in” – the words of “takes place” have been replaced with “is located”

L 115. “some stages”, some is suggested to be replaced by specific number – the word of “some” has been replaced with “five” 

Minor improvements can still be made in the Result section.

Figure 3 requires some improvement. The legends are hardly readable, in 3a two spots of purple areas do not correspond to some symbol from the legend, 3b is very dense in some parts and the patterns used do not help readability. – done with additional explanation in the cover letter

L 206 “landuse”, please correct in land use - done

L220 “soil erosions”, please correct in soil erosion - done

L221 “trees barriers”, please correct into tree barriers - done